# FAKEXPLAIN: AI-GENERATED IMAGE DETECTION VIA HUMAN-ALIGNED GROUNDED REASONING

**Yikun Ji**[1,2]   **Yan Hong**[2]   **Qi Fan**[1]   **Jun Lan**[†2]
**Huijia Zhu**[2]   **Weiqiang Wang**[2]   **Liqing Zhang**[†1]   **Jianfu Zhang**[†1]
[1]Shanghai Jiao Tong University   [2]Ant Group
{da-kun, lqzhang, c.sis}@sjtu.edu.cn, yelan.lj@antgroup.com [*]

## ABSTRACT

The rapid rise of image generation calls for detection methods that are both interpretable and reliable. Existing approaches, though accurate, act as black boxes and fail to generalize to out-of-distribution data, while multi-modal large language models (MLLMs) provide reasoning ability but often hallucinate. To address these issues, we construct **FakeXplained** dataset of AI-generated images annotated with bounding boxes and descriptive captions that highlight synthesis artifacts, forming the basis for human-aligned, visually grounded reasoning. Leveraging **FakeXplained**, we develop **FakeXplainer** which fine-tunes MLLMs with a progressive training pipeline, enabling accurate detection, artifact localization, and coherent textual explanations. Extensive experiments show that **FakeXplainer** not only sets a new state-of-the-art in detection and localization accuracy (98.2% accuracy, 36.0% IoU), but also demonstrates strong robustness and out-of-distribution generalization, uniquely delivering spatially grounded, human-aligned rationales. The code and dataset are available at: https://github.com/Gennadiyev/FakeXplain.

## 1 INTRODUCTION

The past decade has witnessed rapid progress in text-to-image generation, evolving from Generative Adversarial Networks to Diffusion Models, which are now capable of producing images nearly indistinguishable from real photographs (Goodfellow et al., 2014; Peebles & Xie, 2023). These advances have led to an explosion of highly realistic AI-generated content, raising pressing concerns about misinformation, authenticity, and trust in digital media. Most existing detection methods cast this task as a binary classification problem, leveraging convolutional neural networks or vision transformers (Wang et al., 2020; Ojha et al., 2023; Park & Owens, 2024). However, binary labels offer limited insight into *why* an image is classified as AI-generated. In real-world applications, especially those involving legal, journalistic, or ethical implications, explainable detection is essential. An effective detection system should not only identify whether an image is AI-generated but also pinpoint the specific visual cues or logical inconsistencies that betray its synthetic origin. Such explainability promotes user trust, supports verification workflows, and enables more informed decision-making.

The rise of Multi-modal Large Language Models (MLLMs) has enabled cross-modal inference, allowing models to generate human-readable explanations about AI-generated images. Recent efforts (Li et al., 2024; Zhang et al., 2024; Zhou et al., 2025; Gao et al., 2025; Xu et al., 2024; Liu et al., 2024; Ji et al., 2025) have advanced interpretable textual reasoning using MLLMs. However, these methods either depend heavily on prompt engineering, model-generated explanations, or plug-in segmentation modules (Kirillov et al., 2023) to delineate manipulated regions. As illustrated in Figure 1, existing MLLM-based detectors may hallucinate false claims or provide reasons without spatial grounding, since their explanations are not validated by human annotations. Without proper visual grounding or human-aligned supervision, it remains unclear whether the generated rationales truly reflect the image content or derive from model hallucinations. To improve human alignment, fine-grained multimodal supervision, such as region-level annotations and captions, is essential. Yet, the lack of such high-quality datasets poses a major challenge to building reliable and interpretable MLLM-based detection systems. In this paper, we present **FakeXplained**, a dataset of high-quality

---

[*]† Corresponding authors.

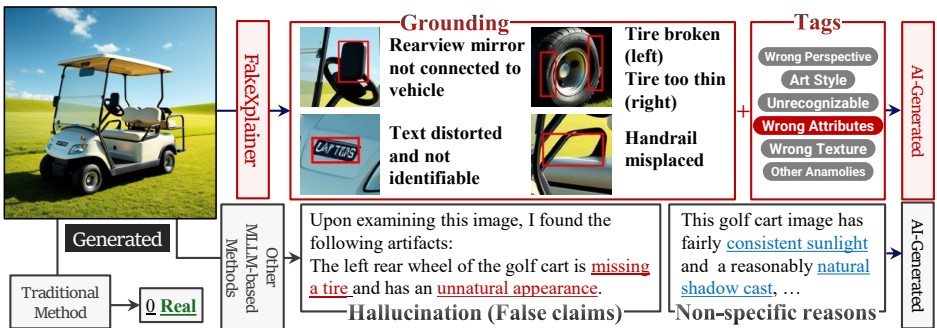

Figure 1: Comparison of our method (**FakeXplainer**) with traditional classification-based detectors (without explanations) and other MLLM-based methods (with hallucinated or non-specific reasons). **FakeXplainer** is trained to localize flawed regions and explain why the image appears AI-generated.

AI-generated images with fine-grained, human-grounded annotations, together with **FakeXplainer**, an RL fine-tuning pipeline for MLLMs that achieves state-of-the-art detection accuracy and grounding performance. As shown in Figure 1, training on FakeXplained enables FakeXplainer to provide comprehensive rationales for fake image detection, performing on par with human experts. Our major contributions are threefold:

- **The FakeXplained dataset** is a curated dataset of 8,772 AI-generated images from diverse state-of-the-art generative models, annotated with bounding boxes and concise captions that highlight visual anomalies and illogical details.

- **The FakeXplainer detector:** By fine-tuning MLLMs on **FakeXplained**, we build an end-to-end system that not only detects AI-generated images but also explains them. Fine-tuning on FakeXplained enables **FakeXplainer** to perform fine-grained visual reasoning and articulate clear, human-aligned observations.

- **Robust performance with explainability:** **FakeXplainer** answers *"where and why does this image look fake?"* with reliable, spatially grounded explanations. Extensive experiments show that it achieves state-of-the-art detection accuracy, generalizes well to out-of-distribution images, and remains robust under perturbations while providing human-aligned, interpretable reasoning.

## 2 RELATED WORKS

**Detection of AI-generated and manipulated images.** Detecting AI-generated images has gained prominence with the improving fidelity of synthetic images from GANs (Goodfellow et al., 2014; Esser et al., 2021), autoregressive transformers (Van Den Oord et al., 2017), diffusion-based models (Le et al., 2025; Ye et al., 2025; Wang et al., 2025b; Li et al., 2025a; Chadebec et al., 2025; Song et al., 2020; Ho et al., 2020) and DiTs (Peebles & Xie, 2023; Chen et al., 2023). Deep learning methods such as ResNets and Vision Transformers trained on real and synthetic data (Wang et al., 2020; Tan & Le, 2019; Park & Owens, 2024; Chang et al., 2023; Yan et al., 2024) leverage strong feature extraction to learn discriminative patterns. However, generalization to unseen models remains challenging (Bi et al., 2023; Yan et al., 2024). As generation techniques evolve, artifact-based cues alone become increasingly unreliable. A complementary research direction focuses on model explainability, as most detectors offer only binary classification without indicating how or where synthetic cues are found. Recent efforts towards fine-grained or localized detection include using multi-branch systems for multi-level labels (Bi et al., 2023), computing local intrinsic dimensionalities (Lorenz et al., 2023), or using gradient visualizations (Selvaraju et al., 2017). Despite these advances, existing MLLM-based methods still face limitations in providing human-aligned, grounded explanations and in generalizing across rapidly evolving generative techniques. While several recent datasets leverage MLLMs to generate rationales or global labels (Zhang et al., 2024; Zhou et al., 2025; Huang et al., 2025b; Wen et al., 2025; Li et al., 2025b), their reliance on automatic annotations raises concerns about hallucination and weak visual grounding. Although FakeBench (Li et al., 2024) incorporates human refinement, its explanations are initially generated by GPT-4V

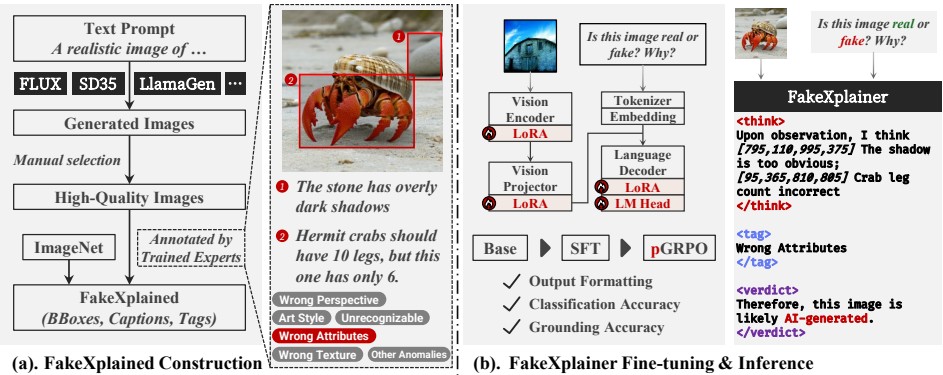

Figure 2: Overview of: **(a) FakeXplained**: Construction of dataset with human-aligned bounding boxes and captions, and **(b) FakeXplainer**: Progressive fine-tuning pipeline of MLLMs, which integrates SFT and GRPO to achieve accurate detection, grounding, and interpretable reasoning.

and remain purely textual, without any region-level grounding; moreover, the full annotations are not publicly released. LOKI covers multiple modalities but relies on GPT-4o for fine-grained labels, which weakens its alignment with humans. Other datasets such as So-Fake-Set (Huang et al., 2025b) focuses on social-media scenarios and similarly lack region-level annotations, limiting their suitability for training grounded, human-aligned detectors. Methodologically, recent detectors such as AIGI-Holmes (Zhou et al., 2025) combine NPR with MLLMs to improve interpretability, whereas localization-focused approaches like FakeShield (Xu et al., 2024) and LEGION (Kang et al., 2025) depend on external segmentation modules (*e.g.*, SAM (Kirillov et al., 2023)), leaving the intrinsic grounding capabilities of MLLMs underutilized. Meanwhile, methods without additional modules, such as So-Fake (Huang et al., 2025b) and FakeVLM (Wen et al., 2025), still cannot generate spatially localized explanations. The absence of high-quality datasets remains a key obstacle to building reliable and interpretable MLLM-based detection systems. Without proper grounding or human-aligned supervision, it is unclear whether generated rationales truly reflect image content or are derived from hallucinations.

**Training & fine-tuning reasoning-capable MLLMs.** Enhancing the reasoning capabilities of MLLMs is crucial for tasks requiring nuanced understanding (Wu et al., 2025a;b; Yang et al., 2025a; Fang et al., 2025; Chen et al., 2024c). Initial strategies involved converting images into formalized textual representations to enable structured, language-driven reasoning (Yang et al., 2025b). Subsequent research has focused on instilling deeper cognitive abilities, including self-verification, self-correction, developing "slow thinking" capabilities (Wang et al., 2025a), and managing reasoning depth to address phenomena like "overthinking" (Xiao et al., 2025). Efforts also explore constructing high-quality multi-modal Chain-of-Thought (CoT) datasets (Huang et al., 2025a) to guide reasoning processes. Reinforcement Learning (RL) has become pivotal in these advancements, with many sophisticated reasoning developments relying on RL methodologies. Fine-tuning methods have spurred significant interest in RL-based multi-modal reasoning (Chen et al., 2025). RL, particularly when combined with structured reward functions, *e.g.*, using Intersection over Union (IoU) for tasks involving image grounding (Shen et al., 2025) - markedly improves multi-modal alignment, visual reasoning, and human interpretable decision making, demonstrating RL's capability of advancing model performance in complex vision-language tasks.

## 3 THE FAKEXPLAINED DATASET

Our objective is trustworthy and interpretable detection of AI-generated images. This requires detectors that generalize to unseen generative models, remain robust to perturbations, and provide human-understandable explanations. Conventional detectors often lack interpretability, while MLLMs, though promising, tend to produce unreliable explanations with frequent hallucinations when used without training (see Table 4). To address this, we require models that not only detect AI-generated images but also explain their decisions in natural language for reliability. Achieving this demands a dataset that supports both visual grounding and textual reasoning. Therefore, we introduce the

**FakeXplained** dataset, as illustrated in Figure 2(a), to train an MLLM to produce trustworthy explanations. It consists of high-quality synthetic images paired with fine-grained, human annotations that indicate the underlying flaws and artifacts responsible for detection as fake.

## 3.1 AI-GENERATED IMAGES SELECTION

**Semantic alignment.** To ensure content diversity and semantic alignment, we generated images (a) across 1,000 ImageNet categories and (b) based on MS COCO (Chen et al., 2015) captions (2017 split). For ImageNet-1K classes, the prompt used is:

*"A realistic image of {class_name}"*

**Model variety.** We ensure source diversity by using 28 text-to-image generation models across architectures. The models used are listed as follows:

**Diffusion-Based Generators:** Midjourney Midjourney (2023), Stable Diffusion models Rombach et al. (2022b); Esser et al. (2024), DDIM Song et al. (2020), DDPM Ho et al. (2020), DALL·E OpenAI (2023), GLIDE Nichol et al. (2021), and VQDM Gu et al. (2022).

**GAN-Based Generators:** GALIP Tao et al. (2023), StyleGAN Karras et al. (2018), VQGAN Esser et al. (2021), and BigGAN Brock et al. (2018).

**DiT-Based Generators:** PixArt Chen et al. (2023; 2024b;a) and DiT Peebles & Xie (2023).

**Other Generators:** VAR Tian et al. (2024), Infinity Han et al. (2024), MaskGIT Chang et al. (2022), and LlamaGen Sun et al. (2024).

Most generated images are at a resolution of $512 \times 512$. For methods that do not natively support this resolution, the $1024 \times 1024$ resolution is used, and the images are downscaled to $512 \times 512$ before entering annotation stage.

**Quality assurance.** All generated images underwent manual quality screening. In this stage, low-quality images that depict non-identifiable objects, or images where no part is real, are removed from the dataset. After screening, 8,772 AI-generated images were selected for subsequent annotation.

## 3.2 IMAGE ANNOTATION

To support interpretable reasoning and help models understand what constitutes an AI-generated image, we provide detailed annotations for synthetic images. Real images are not annotated because they lack synthesis flaws.

**Flawed regions and explanations.** Precise regional annotations and corresponding textual descriptions are essential for visual grounding and interpretability. We recruited 23 trained annotators to label the high-quality AI-generated images selected from the previous stage. Their primary task was to identify and describe all regions within each image that exhibited signs of being fake (detailed guidelines provided in Appendix A.1). Prior to annotation, all participants underwent standardized training focused on identifying visual cues of AI-generated content. The training emphasized the identification of *fake regions*, which are defined as areas within an image that either violate common sense or exhibit noticeable AI-generated artifacts. Examples of common sense violations include anomalies such as "a flamingo with three legs" or "bird feathers with a metallic appearance". Common AIGC artifacts include "repetitive patterns on a blanket" or "blurred or illegible text".

Annotators were also introduced to a structured annotation rubric to ensure consistency and alignment with the dataset's objectives. Each annotation consists of one or more fake regions, where each region is represented by a tuple $(R_i, T_i)$, where $R_i$ denotes a rectangular bounding box encapsulating the region, and $T_i$ provides a textual description of the identified anomaly or artifact. On average, an annotated image in the dataset contains **5.42** such $(R_i, T_i)$ pairs, which serve as the foundation for grounding and reasoning in downstream model training.

**Image-level tagging.** In addition to region-level annotation, annotators were asked to tag images based on broader perceptual attributes. These attributes include texture quality, overall realism, correctness of attributes, recognizability of objects, and the presence of other significant defects not explicitly listed (*e.g.*, the occurrence of multiple sub-images within a single image). These tags $C_i$

are mutually independent, allowing annotators to assign zero or multiple tags to each image as appropriate. This tagging framework allows the dataset to capture holistic image quality assessments, particularly in cases where visually realistic AI-generated images may lack distinct localized flaws.

## 3.3 QUALITY CONTROL

To ensure the reliability of the annotations, we implemented a quality control protocol involving both manual inspection and algorithmic validation. A subset of annotations was compared against a reference set of fake region annotations curated by the research team. Given the inherently subjective nature of visual interpretation, we adopted a tolerant validation criterion to accommodate diverse perspectives among annotators. Specifically, a minimum Intersection over Union (IoU) threshold of 20.0% was applied for bounding box overlap, and an accuracy threshold of $\frac{1}{3}$ was used for image-level tagging. These metrics were assessed on a validation set comprising 5% of the annotated images. The IoU metric is used to assess the spatial agreement between annotated and reference bounding boxes. Let $R_v$ represent the rectangular bounding box annotated by a volunteer, and $R_r$ represent the corresponding reference bounding box from the reference set. The Intersection over Union (IoU) is computed as:

$$\text{IoU}(R_v, R_r) = \frac{|R_v \cap R_r|}{|R_v \cup R_r|},$$

where $|R_v \cap R_r|$ denotes the area of the intersection between $R_v$ and $R_r$, and $|R_v \cup R_r|$ denotes the area of their union. The IoU value ranges from 0 to 1, with higher values indicating stronger alignment. This quality control procedure ensures a baseline level of annotation fidelity while preserving the diversity of human interpretations. The resulting dataset, enriched with both region-level annotations $(R, T)$ and image-level tags $C$, offers a robust foundation for analyzing the semantic inconsistencies and perceptual flaws of AI-generated images.

## 4 METHODOLOGY: FAKEXPLAINER

We propose a training methodology named **FakeXplainer** for MLLMs designed to detect AI-generated imagery, localize relevant artifacts, and articulate the rationale for their predictions. The training and inference pipeline is shown in Figure 2(b). Inspired by DeepSeek-Math (Shao et al., 2024), the training pipeline begins with Supervised Fine-Tuning (SFT) (Ouyang et al., 2022) to provide basic reasoning ability and ensure structured outputs. This initial phase is succeeded by Reinforcement Learning from Human Feedback (RLHF), which is implemented using *progressive* GRPO (pGRPO), leveraging our proposed FakeXplained dataset.

Before training, each image's annotations are reformatted into a dialogue between a user and an assistant, using a prompt structure designed for localization-aware fine-tuning. Region-level annotations $(R_i, T_i)$ are enclosed within `<think>` markers, image-level tags $C_i$ within `<tag>` markers, and the final verdict is wrapped in `<verdict>` markers.

### 4.1 COLD START WITH SUPERVISED FINE-TUNING

The cold start phase of **FakeXplainer** uses SFT to establish a stable foundation before proceeding to RL. During this phase, all linear layers of the vision encoder, projector, and language model components in the MLLM are fine-tuned based on the supervision signals from the data. This initial fine-tuning is crucial for stabilizing the model prior to full-scale reinforcement learning training, preventing instabilities that might arise from pure RL-based updates (Guo et al., 2025).

The SFT process focuses on teaching the model to produce coherent reasoning patterns with a clear structure. The training emphasizes the consistent use of the designated marker format with `<think>`, `<tag>`, and `<verdict>` fields, ensuring format clarity in the model's reasoning outputs. This structured Chain-of-Thought (CoT) format reduces errors and improves explainability, providing a solid foundation for subsequent GRPO stages that will refine the model's performance on specific metrics.

## 4.2 DESIGN OF REWARD FUNCTIONS

Reward design is a critical component of RLHF, guiding the MLLMs to learn not only how to detect fake images, but also how to localize relevant regions and provide coherent reasoning. We define three core reward functions for this purpose.

**Classification accuracy (*Label*).** To ensure the model produces the correct verdict, we extract the classification decision from within the `verdict` marker and compare it with the ground-truth label. Let $o$ denote the textual output of the MLLM, we have:

$$\mathcal{R}_C(o) = \begin{cases} 1, & \text{if } V(o) = y, \\ 0, & \text{o.w.} \end{cases}$$

where $V(o)$ is a regex match for the verdict, and $y$ is the ground-truth label of whether the image is real or generated.

**Grounding accuracy (*IoU*).** To reward alignment between model-predicted and human-annotated regions, we use a relaxed version of the Intersection over Union (IoU):

$$\mathcal{R}_G(o) = \text{IoU}^{\times \eta} = \min\left(1, \eta \, \text{IoU}(R(o), R_y)\right)$$

where $R(o)$ is the region extraction function that parses textual output $o$ to bounding boxes, $R_y$ is the annotated region, and $\eta$ is a relaxing constant. The relaxation is based on the observation that human annotators have slight discrepancies regarding the borders of annotated regions. This relaxation reward ensures full credit to the model when the regions annotated by the model are in good correlation with human-annotated ones.

**Output format validity (*Format*).** To ensure the model understands the structural requirements of the task, we introduce a format reward that encourages outputs conforming to the expected syntax. A valid output must include correctly structured `<think>`, `<tag>` and `<verdict>` markers, as well as bounding boxes and captions that are syntactically well-formed and can be parsed using regular expressions. Formally, the reward is defined as:

$$\mathcal{R}_F(o) = \begin{cases} 1, & \text{if } \{V, R, T, C\}(o) \text{ are parsable} \\ 0, & \text{o.w.} \end{cases}$$

where $T(o)$ and $C(o)$ extract the regional captions and image-level tags from $o$, respectively.

## 4.3 RLHF WITH GROUP RELATIVE POLICY OPTIMIZATION

Following SFT, we employ Group Relative Policy Optimization (GRPO) Shao et al. (2024) to progressively align the MLLM with our objectives of interpretable and reliable fake image detection. GRPO combines structured supervision from the dataset with targeted reward signals through a carefully designed training process. The reward function is formulated as:

$$\mathcal{R} = \omega_G(t)\mathcal{R}_G + \omega_C\mathcal{R}_C + \omega_F\mathcal{R}_F, \tag{1}$$

where weights $\omega_C = \omega_F = 1.0$ remain constant throughout training, while $\omega_G(t)$ increase linearly from 0.5 to 1.0 over the training process.

Our approach employs continuous linear interpolation for the localization weight:

$$\omega_G(t) = 0.5 + 0.5 \cdot (t/T), \tag{2}$$

where $t$ represents the current training step and $T$ denotes the total number of training steps.

The linear weighting strategy addresses challenges observed in preliminary experiments and offers three benefits. First, without IoU weight adjustment, models trained with equal weights from the start tend to over-optimize localization rewards, producing many small fragmented bounding boxes that achieve high IoU scores but fail to capture meaningful regions. Down-weighting the localization reward in early training prevents this issue. Second, the progressive scheme enables a natural curriculum learning. With reduced localization weight at the beginning, the model first learns output formatting and classification accuracy. As these skills stabilize, the gradually increasing IoU reward improves localization on top of this foundation. Third, continuous weight adjustment avoids reward spikes and stabilizes training, allowing smooth adaptation of optimization objectives. Experiments show that linear reward weighting outperforms static schemes, confirming the effectiveness of gradual reward shaping for training MLLMs in complex visual understanding tasks.

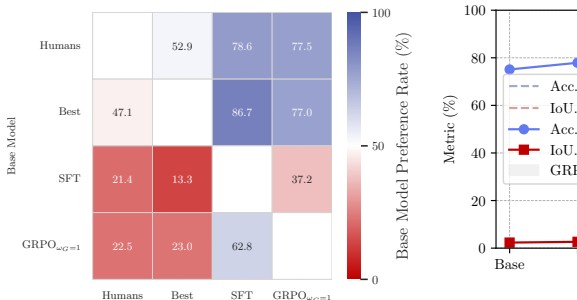

Figure 3: Human preference matrix.

Figure 4: Accuracy and IoU curve of the FakeX-plainer during the training process.

## 5 EXPERIMENTS

### 5.1 EXPERIMENTAL SETUP

We adopt *Qwen2.5-VL-Instruct* (Qwen Team, 2025) as the base model of our FakeXplainer for its strong pre-trained grounding capabilities. Both the SFT and GRPO stages last for three epochs, with a batch size of 1. We use $\eta = 1.1$ and set the number of samples to $G = 4$ during the GRPO stage. We use 8x NVIDIA A100 GPUs in the SFT stage; 16x NVIDIA A100 GPUs in the GRPO stage. All baseline methods are trained on one NVIDIA A100 GPU. More experimental details are provided in the Appendix B. For baseline comparisons, we use the same training data as the Fak-eXplainer setup. We train SegFormer (Xie et al., 2021) and ObjectFormer (Wang et al., 2022) under a segmentation + classification setting on the FakeXplained dataset by converting bounding boxes to binary masks. For classification-only methods, including NPR (Ojha et al., 2023), DMD (Corvi et al., 2023), ComFor (Park & Owens, 2024), AfPr (Chang et al., 2023), and DIRE (Wang et al., 2023), only image-level labels are used during training. We additionally evaluated state-of-the-art MLLM-based detection methods, specifically FakeShield (Xu et al., 2024), LEGION (Kang et al., 2025), and FakeVLM (Wen et al., 2025). For FakeShield, we utilized the pre-trained weights provided by the authors without further fine-tuning. For LEGION, we adhered to the training protocol specified in the original paper, training the model on the SynthScars dataset with identical hyperparameters and experimental configurations as reported.

### 5.2 OVERALL PERFORMANCE

To ensure robustness and mitigate dataset bias, all models are evaluated using four-fold cross-validation. During training, the detection model is exposed to 75% of the images from the Fak-eXplained dataset along with an equal number of real samples. Evaluation is conducted on the remaining 25% of synthetic images, again paired with the same number of real images. We report both classification accuracy and localization performance using the IoU metric on AI-generated images. Robustness tests against perturbations are provided in Appendix C.

**Comparing to other methods.** Quantitative results are reported in Table 1, comparing **FakeX-plainer** with traditional detectors. For MLLMs, Table 2 presents the post-finetuning performance of LEGION (Kang et al., 2025) and FakeShield (Xu et al., 2024) across different pre-trained models. Our best-performing model achieves an overall classification accuracy of **98.2%**, demonstrating strong robustness and consistent performance across different image generators. For localization, the model achieves an IoU score of **36.0%**, outperforming all segmentation-based baselines. This indicates that **FakeXplainer** identifies fake regions more consistently with human annotations than competing approaches. **Reasoning Quality.** Table 2 shows the BLEU-2 and ROUGE-L (Li et al., 2024) of model responses against the FakeXplained dataset. The results verify the training effectiveness of FakeXplainer, considerably outperforming the base model in explanation generation, indicating that both the regions and their reasons are generated accurately.

**Generalizability of FakeXplainer on other MLLMs.** To assess generalizability beyond Qwen-2.5-VL, we further evaluated it on several state-of-the-art MLLMs with diverse architectures and capa-

Table 1: Experimental result for current AI-generated image detectors and our FakeXplainer across different image generation methods.

| Generators | FakeXplainer | | ObjectFormer | | SegFormer | | NPR | DMD. | ComFor. | AfPr. | AEROB. | DIRE | FakeVLM |
|---|---|---|---|---|---|---|---|---|---|---|---|---|---|
| | Acc. | IoU | Acc. | IoU | Acc. | IoU | Acc. | Acc. | Acc. | Acc. | Acc. | Acc. | Acc. |
| DALL·E 2 2022 | **0.986** | 0.360 | 0.957 | 0.251 | 0.942 | 0.285 | 0.907 | 0.934 | 0.877 | 0.892 | 0.823 | 0.916 | 0.908 |
| DALL·E 3 2023 | **0.991** | 0.365 | 0.949 | 0.258 | 0.950 | 0.292 | 0.912 | 0.942 | 0.872 | 0.907 | 0.821 | 0.923 | 0.915 |
| DDIM 2020 | **0.974** | 0.345 | 0.954 | 0.285 | 0.945 | 0.280 | 0.917 | 0.928 | 0.879 | 0.915 | 0.839 | 0.912 | 0.902 |
| DDPM 2020 | **0.979** | 0.350 | 0.951 | 0.293 | 0.947 | 0.288 | 0.903 | 0.931 | 0.876 | 0.898 | 0.836 | 0.917 | 0.906 |
| FLUX.1-dev 2024 | **0.988** | 0.362 | 0.958 | 0.299 | 0.940 | 0.295 | 0.922 | 0.937 | 0.874 | 0.779 | 0.843 | 0.919 | 0.938 |
| FLUX.1-schnell | **0.972** | 0.343 | 0.953 | 0.287 | 0.943 | 0.283 | 0.926 | 0.929 | 0.882 | 0.805 | 0.827 | 0.913 | 0.941 |
| GLIDE 2021 | **0.970** | 0.340 | 0.950 | 0.289 | 0.946 | 0.286 | 0.913 | 0.935 | 0.873 | 0.661 | 0.822 | 0.922 | 0.897 |
| Midjourney v4 2023 | **0.990** | 0.364 | 0.956 | 0.296 | 0.949 | 0.294 | 0.908 | 0.939 | 0.869 | 0.878 | 0.814 | 0.925 | 0.926 |
| Midjourney v5 | **0.992** | 0.366 | 0.959 | 0.273 | 0.941 | 0.297 | 0.902 | 0.943 | 0.871 | 0.851 | 0.718 | 0.927 | 0.932 |
| SD 1.4 2022a | 0.968 | 0.338 | 0.952 | 0.286 | 0.944 | 0.282 | 0.921 | **0.970** | 0.880 | 0.852 | 0.951 | 0.909 | 0.918 |
| SD 1.5 | **0.975** | 0.347 | 0.955 | 0.294 | 0.948 | 0.290 | 0.916 | 0.949 | 0.875 | 0.866 | 0.966 | 0.915 | 0.921 |
| SD 2.1 2022b | **0.980** | 0.352 | 0.951 | 0.291 | 0.942 | 0.287 | 0.911 | 0.938 | 0.872 | 0.881 | 0.833 | 0.918 | 0.913 |
| SD 3.5 Large 2024 | **0.991** | 0.365 | 0.954 | 0.294 | 0.945 | 0.293 | 0.904 | 0.944 | 0.870 | 0.934 | 0.830 | 0.924 | 0.929 |
| SD 3.5 Large Turbo | **0.993** | 0.368 | 0.957 | 0.312 | 0.950 | 0.296 | 0.906 | 0.947 | 0.868 | 0.927 | 0.837 | 0.928 | 0.935 |
| VQDM 2022 | **0.973** | 0.342 | 0.953 | 0.288 | 0.943 | 0.284 | 0.927 | 0.932 | 0.877 | 0.938 | 0.932 | 0.914 | 0.909 |
| *Diffusion* | *0.983* | *0.356* | *0.954* | *0.287* | *0.945* | *0.290* | *0.913* | *0.941* | *0.874* | *0.864* | *0.842* | *0.920* | *0.919* |
| BigGAN 2018 | **0.965** | 0.335 | 0.950 | 0.280 | 0.941 | 0.278 | 0.918 | 0.892 | 0.903 | 0.933 | 0.861 | 0.887 | 0.894 |
| GALIP 2023 | 0.882 | 0.353 | **0.941** | 0.279 | **0.941** | 0.289 | 0.882 | 0.882 | 0.941 | 0.353 | 0.706 | 0.882 | 0.876 |
| VQGAN 2021 | **0.967** | 0.337 | 0.954 | 0.282 | 0.943 | 0.280 | 0.907 | 0.889 | 0.908 | 0.921 | 0.932 | 0.885 | 0.858 |
| StyleGAN-XL 2018 | 0.960 | 0.330 | 0.951 | 0.278 | 0.940 | 0.275 | 0.914 | 0.884 | **0.980** | 0.928 | 0.939 | 0.879 | 0.680 |
| *GAN* | *0.955* | *0.337* | *0.950* | *0.280* | *0.941* | *0.279* | *0.912* | *0.890* | *0.916* | *0.866* | *0.860* | *0.885* | *0.827* |
| PixArtAlpha 2023 | **0.987** | 0.357 | 0.956 | 0.295 | 0.947 | 0.291 | 0.908 | 0.912 | 0.891 | 0.934 | 0.927 | 0.903 | 0.861 |
| PixArtDelta 2024b | **0.984** | 0.354 | 0.953 | 0.292 | 0.943 | 0.289 | 0.921 | 0.909 | 0.893 | 0.939 | 0.922 | 0.899 | 0.896 |
| PixArtSigma 2024a | **0.989** | 0.360 | 0.957 | 0.296 | 0.949 | 0.293 | 0.919 | 0.915 | 0.889 | 0.924 | 0.931 | 0.905 | 0.904 |
| DiT 2023 | **0.978** | 0.349 | 0.952 | 0.290 | 0.942 | 0.287 | 0.913 | 0.907 | 0.896 | 0.928 | 0.938 | 0.897 | 0.892 |
| *DiT* | *0.983* | *0.354* | *0.954* | *0.293* | *0.945* | *0.289* | *0.914* | *0.910* | *0.893* | *0.931* | *0.931* | *0.900* | *0.889* |
| VAR 2024 | **0.976** | 0.346 | 0.954 | 0.287 | 0.945 | 0.283 | 0.928 | 0.893 | 0.901 | 0.934 | 0.927 | 0.889 | 0.886 |
| Infinity 2024 | **0.974** | 0.344 | 0.951 | 0.289 | 0.941 | 0.286 | 0.914 | 0.897 | 0.899 | 0.938 | 0.924 | 0.883 | 0.872 |
| MaskGIT 2022 | **0.972** | 0.342 | 0.955 | 0.288 | 0.948 | 0.284 | 0.909 | 0.895 | 0.904 | 0.923 | 0.933 | 0.886 | 0.854 |
| LlamaGen 2024 | **0.980** | 0.351 | 0.953 | 0.429 | 0.944 | 0.289 | 0.923 | 0.899 | 0.897 | 0.929 | 0.938 | 0.892 | 0.867 |
| *Others* | *0.978* | *0.348* | *0.953* | *0.369* | *0.944* | *0.287* | *0.920* | *0.898* | *0.898* | *0.931* | *0.933* | *0.889* | *0.870* |
| Real Images 2009 | **0.985** | - | 0.956 | - | 0.946 | - | 0.918 | 0.903 | 0.882 | 0.934 | 0.854 | 0.896 | 0.763 |
| Overall | **0.982** | 0.360 | 0.954 | 0.299 | 0.945 | 0.289 | 0.914 | 0.928 | 0.882 | 0.887 | 0.873 | 0.911 | 0.828 |

Table 2: Performance comparison of FakeXplainer across different base MLLMs and against other MLLM-based methods. Post-finetuning results are underlined.

| Method | FakeXplainer | | | | | | | | | | FakeShield | LEGION |
|---|---|---|---|---|---|---|---|---|---|---|---|---|
| Backbone | InternVL3-8B | | InternVL3-14B | | Ovis2.5-9B | | MiMo-VL-7B-RL | | **Qwen-2.5-VL-32B** | | | |
| Acc. | 0.584 | 0.928 | 0.568 | 0.951 | 0.624 | 0.909 | 0.515 | 0.920 | **0.734** | **0.982** | 0.801 | 0.583 |
| IoU. | 0.039 | 0.134 | 0.043 | 0.289 | - | - | - | - | 0.044 | **0.360** | 0.028 | 0.098 |
| BLEU-2 | 0.061 | 0.232 | 0.098 | 0.235 | 0.058 | 0.203 | 0.083 | 0.249 | **0.080** | **0.267** | 0.004 | 0.072 |
| ROUGE-L | 0.059 | 0.225 | 0.092 | 0.219 | 0.050 | 0.184 | 0.076 | 0.239 | **0.076** | **0.251** | 0.003 | 0.055 |

bilities in Table 2. The consistent performance gain of FakeXplainer across architectures, whether with grounding capabilities (InternVL3 (Zhu et al., 2025)) or without (Ovis2.5 (Lu et al., 2024), MiMo (Xiaomi, 2025)), validates the model-agnostic nature of our pipeline.

**Out-of-distribution (OoD) evaluation.** We also evaluated the models on five OoD datasets, Fake-Clue (Wen et al., 2025), Chameleon (Yan et al., 2024), FaceForensics++ (Rössler et al., 2019), images generated by GPT-Image-1 (OpenAI, 2025; Rapidata, 2025) and MMFR-Dataset (eval) proposed by FakeReasoning (Gao et al., 2025). As shown in Table 3, our model consistently outperforms all other methods across OoD datasets, demonstrating considerable generalization to unseen image domains.

Table 3: Accuracy on external datasets for out-of-distribution generalization testing.

| Sources | **FXP.** | ObjFormer. | SegFormer | NPR | DMD. | ComFor. | AfPr. | AEROB. | DIRE | FakeShield | LEGION |
|---|---|---|---|---|---|---|---|---|---|---|---|
| FakeClue 2025 | **0.852** | 0.462 | 0.485 | 0.833 | 0.734 | 0.766 | 0.849 | 0.239 | 0.727 | 0.550 | 0.172 |
| Chameleon 2024 | **0.843** | 0.485 | 0.508 | 0.794 | 0.721 | 0.757 | 0.803 | 0.291 | 0.752 | 0.587 | 0.197 |
| GPT-Image-1 2025 | **0.801** | 0.513 | 0.538 | 0.790 | 0.735 | 0.636 | 0.597 | 0.458 | 0.793 | 0.752 | 0.238 |
| FaceForensics++ 2019 | **0.864** | 0.598 | 0.716 | 0.861 | 0.562 | 0.429 | 0.746 | 0.681 | 0.850 | 0.773 | 0.395 |
| MMFR-Dataset 2025 | **0.874** | 0.653 | 0.657 | 0.569 | 0.619 | 0.595 | 0.786 | 0.685 | 0.624 | 0.710 | 0.193 |

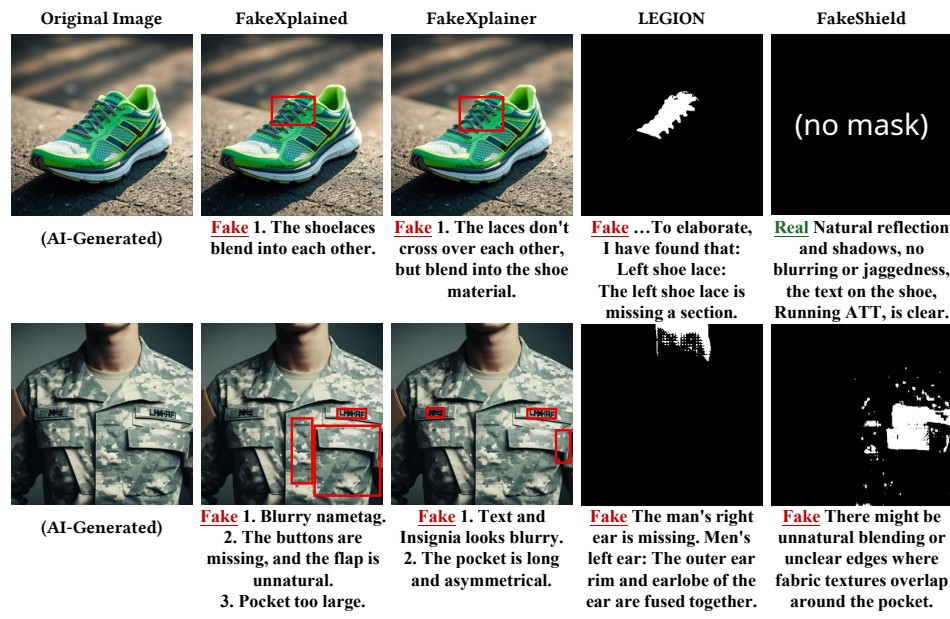

Figure 5: Comparison of responses visualized from the **FakeXplainer** method, the ground truth from the **FakeXplained** dataset, and LEGION (Kang et al., 2025) and FakeShield (Xu et al., 2024).

Table 4: Performance comparison of FakeXplainer under different training configurations.

| Metric | FakeXplainer | | | No-FT | | Training Strategy (32B) | | Partial Data (32B) | | | |
|---|---|---|---|---|---|---|---|---|---|---|---|
| | 3B | 7B | **32B** | 32B | SFT | GRPO$_{\omega_G=1}$ | GRPO$_{\omega_G=0.5}$ | no-bbox | no-caption | no-tags | label-only |
| Acc. | 0.842 | 0.958 | **0.982** | 0.734 | 0.893 | 0.937 | 0.974 | 0.952 | 0.942 | 0.962 | 0.937 |
| IoU. | 0.185 | 0.255 | **0.360** | 0.044 | 0.043 | 0.265 | 0.223 | - | 0.265 | 0.358 | - |
| BLEU-2 | 0.195 | 0.246 | **0.267** | 0.080 | 0.183 | 0.257 | 0.261 | 0.164 | - | 0.243 | - |
| ROUGE-L | 0.121 | 0.218 | **0.251** | 0.076 | 0.174 | 0.239 | 0.242 | 0.160 | - | 0.237 | - |

**Qualitative evaluation.** Figure 5 shows two samples from the FakeXplained test set. We found that our model prefers outlining smaller regions than human annotators, demonstrating fine-grained localization capability. Compared with LEGION and FakeShield, **FakeXplainer** not only provides correct predictions but also delivers reliable, grounded explanations. Our user study also shows that responses from FakeXplainer are preferred 99.75% of the time when compared to LEGION and FakeShield. More qualitative examples will be provided in the Appendix A.2.

**Human preference evaluation.** While IoU and classification accuracy provide objective metrics for detection performance, they do not fully capture the qualitative aspects of region-caption alignment. In fact, the model may, in some instances, generate annotations that surpass those of the original human annotators. To comprehensively assess the quality and relevance of the generated explanations, we conducted a human preference study involving an independent group of evaluators. In this study, participants were shown pairs of outputs for the same image, each with different bounding box annotations and associated captions. With no metadata given, evaluators were asked to choose the annotation that demonstrated better alignment between the region and caption, as well as higher overall quality. If no clear preference emerged, a neutral option was available.

We received 1,525 non-neutral preference votes. In Figure 3, the "Humans" category represents annotations from the FakeXplained dataset. When compared with FakeXplainer, human annotations were preferred in 52.9% of cases, indicating that FakeXplainer achieves near-parity with human annotators in producing region-grounded explanations, demonstrating the effectiveness of our framework in generating high-quality visual-textual reasoning.

### 5.3 ABLATION STUDIES

We ablate each training component, including each data component and each training segment, and report the results in Table 4. Additional ablation results are provided in Appendix D.

**Model size of MLLM.** Model size has a clear impact on the performance of FakeXplainer. With the same training pipeline, the 7B variants can identify the authenticity of the image in 95.8% of the cases, but the 3B variant fails to surpass most traditional methods in detection accuracy and cannot effectively localize fake regions. The 7B variant also outputs a good rationale according to the BLEU and ROUGE-L metrics, making it a good balance between performance and speed.

**Effects of different training stages.** To analyze the impact of each training stage, we report both accuracy and IoU metrics throughout the training process in Figure 4. Without GRPO, SFT alone yields marginal improvements over the base model, especially in localization. The GRPO stage with a constant $\omega_G = 1$ has a higher IoU than the linear scheme but struggles to train effectively in later steps, demonstrating the cumulative benefit of the progressive reward design. By the completion of the RLHF stage, the model reaches an accuracy of 98.2% and an IoU of 36.0%.

**Fine-tuning impact.** Without fine-tuning, Qwen-2.5-VL-32B-Instruct achieves only 73.4% accuracy. SFT improves this to 89.3%, and adding GRPO further increases the performance to 98.2%, demonstrating the critical role of our two-stage training pipeline and the FakeXplained dataset.

**Data components.** We evaluate three data components: image tags, region annotations (bounding boxes + captions), and binary labels. Using only binary labels yields 93.7% accuracy—the lowest among partial variants but still exceeding DMD's 92.8%. Removing bounding boxes or captions reduces accuracy by 3.5%, with caption removal severely impacting IoU (-9.5%). While tag removal has a minimal effect on both metrics. These results confirm that structured reasoning information, particularly region-level annotations, substantially improves detection performance.

**Reward weighting mechanism.** Fixed reward weighting (GRPO$_{\omega_G=1}$ and GRPO$_{\omega_G=0.5}$) underperforms our progressive GRPO approach across all metrics. Notably, the localization-prioritized GRPO$_{\omega_G=1}$ also shows inferior IoU, validating the necessity of textual explanations and dynamic reward weighting for the step-by-step acquisition of classification, localization skills, and overall interpretability.

## 6 CONCLUSION

In this work, we present FakeXplainer, an explainable AI-generated image detection approach utilizing MLLMs that provides grounded, human-interpretable explanations alongside the detection results. The system achieves strong performance metrics (98.2% accuracy, 36.0% IoU) through a progressive training pipeline, establishing a foundation for transparent visual media authentication. Although grounded explainability improves FakeXplainer's generalization, its dependence on human-perceptible artifacts reflects an inherent limitation shared by all current explainable AIGI detectors. Fully realistic synthetic images without semantically describable flaws represent a fundamentally different detection problem and fall outside the scope of this work. At the same time, our results show that grounded, human-aligned reasoning provides clear advantages over black-box classifiers: models that output only a binary label exhibit substantial degradation under distribution shift, whereas FakeXplainer maintains strong performance across multiple out-of-distribution benchmarks. These findings suggest that explicit localization and human-aligned supervision offer a more robust and verifiable signal than classification alone.

## 7 ACKNOWLEDGMENTS

This work was supported in part by the National Natural Science Foundation of China (Grant Nos. 62302295, 62595733, and 62561160155), the Shanghai Municipal Science and Technology Major Project (Grant No. 2021SHZDZX0102). This work was also supported by Ant Group.

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

## A    APPENDIX FOR FAKEXPLAINED

The FakeXplained dataset contains 8,772 high-quality AI-generated images annotated with fine-grained bounding boxes and descriptive captions that highlight synthesis artifacts and logical inconsistencies. This dataset addresses the critical gap in explainable and spatially grounded AI-generated image detection.

### A.1    ANNOTATION PROCESS

We recruited a team of experts who are trained to identify fake regions accurately. All of them have prior experience in photography, have seen AI-generated images before, possess a fundamental photographic literacy understanding, and are familiar with related concepts such as "saturation," "shadow," "perspective," and "noise." During the training process, we provide all annotators with a detailed instruction handout with examples. The handout contains positive and negative examples for each global tag, a detailed bounding box annotation guideline, along with quality control metrics.

**Instructions on fake regions.** The rule for annotating fake regions is, if through observation of the selected regions of interest, humans should be able to clearly determine that the image is not an authentic photograph. Fake regions primarily show objects that do not follow the natural physics laws or contradict common sense. Common image generation artifacts are also encouraged to be annotated. After selecting a local area in the image, it is necessary to describe the reason for identifying it as a generated image. The descriptive sentence must start with a noun, followed by one or several adjective phrases or short clauses, and must exclusively describe content that appears in the region.

**Definition of tags.** We refer to the most prominent depicted object in the non-background portion of the image as the *image subject*. There are exactly six different tags that annotators can attach to an image. Their definitions are listed as follows:

- **Perspective errors:** Indicates that the image has an unnatural viewing angle, or errors in perspective, vanishing points. Incorrect occlusion and shadow errors do not constitute perspective errors, but can be considered as fake regions instead.

- **Artistic styles:** If the overall image presents any artistic style, including but not limited to oil painting, ink painting, or manga style, then select the "Artistic Style" tag. If only a certain part of the image contains content in an artistic style, this tag should not be selected.

- **Unknown objects:** Indicates that the *subject* of the image does not exist in the world, or is obviously unreasonable. There may be unusual insects and furniture with strong design elements. Judgment should be based on intuition; unfamiliar or rare subjects do not necessarily indicate unreasonable or non-existent objects.

- **Structure/attribute errors:** Indicates that the **subject** of the image has a structure that is inconsistent with common knowledge, or has attributes inconsistent with common knowledge. Examples include green flower petals, pink elephants, bent iron spoon handles, humans with more than two legs, and asymmetrical shapes. For erroneous attributes that only occupy a small portion of the image subject, such as an incorrect number of fingers on a human hand, fake regions should be marked as well.

- **Texture errors:** If obvious texture errors appear in the image, this tag needs to be selected. For example, the texture of the entire image is blurry, or a portion of an object has a repetitive, tilted, or distorted texture. Unreadable text does not qualify as a texture error and should be labeled as a fake region instead. If "Artistic Style" has already been marked, this tag is usually omitted.

- **Other anomalies:** If there are very obvious global errors in the image that do not belong to any of the above categories, check this item. This tag can also be marked even if other tags have already been chosen.

**Keywords in Annotations** We analyze the captions of the bounding boxes to find the most frequent phrases. Their occurrences are shown in Figure 6. Since we filter for the highest-quality images, it is hard to find a deciding bounding box for some cases. The contours and depth of field are more likely to give the image away, leading to a high frequency of related captions. FakeXplainer manages to align with most of the FakeXplained traits, with a higher detection rate for abnormal object textures.

**Quality Check** To ensure quality, we cross-referenced the annotators' proposals against control samples we marked ourselves. We enforced a rejection policy where an average IoU under 20% or a tag agreement lower than $\frac{1}{3}$ will result in the exclusion of all previous data provided by the annotator.

Figure 7 qualitatively demonstrates the 20% IoU criteria of the screening process. The criteria are selected to accommodate individual understandings over AI-generated images while preventing obvious fake regions from being omitted without notice. When the annotation process finishes, the annotations and QC samples reached an IoU of 42.35% and the tag-agreement rate was 79.67%.

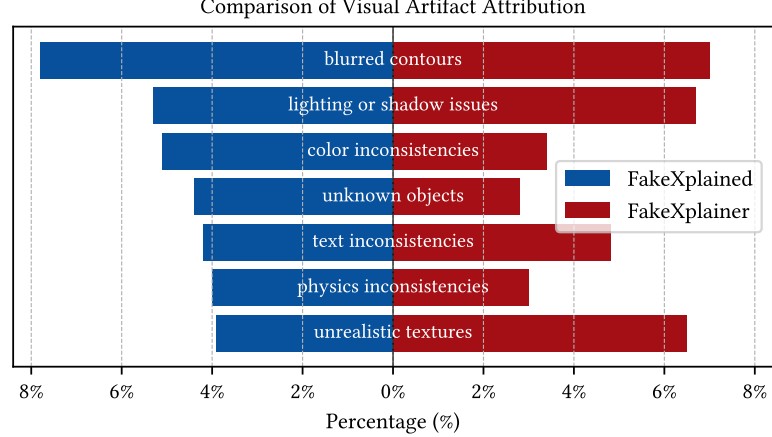

Figure 6: Keyword analysis for FakeXplained and the FakeXplainer responses.

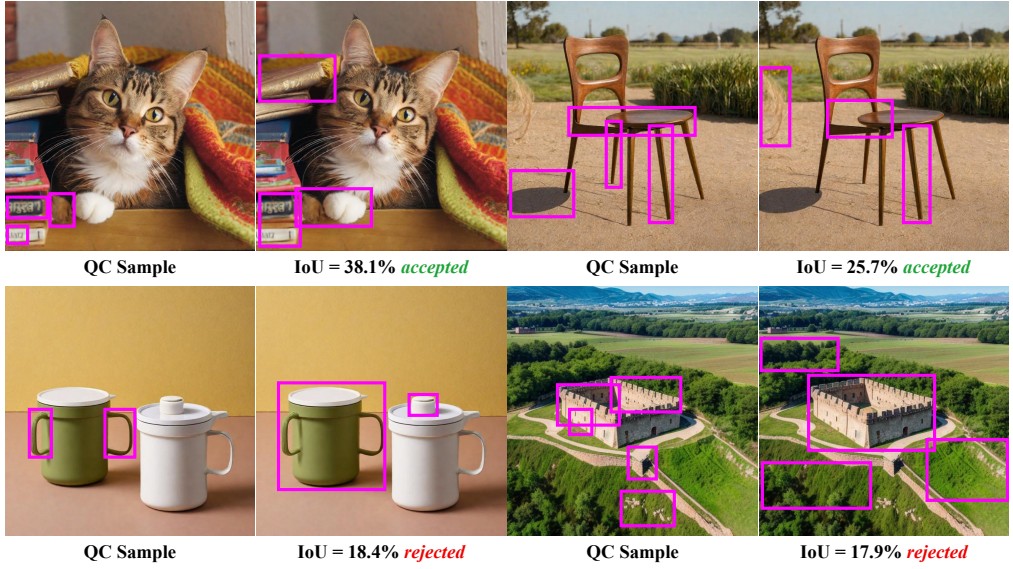

Figure 7: Samples from the quality check process. When our regional annotation has a lower than 20% IoU with the annotation proposal, we reject the annotation.

## A.2 MORE SAMPLES

Figure 9 presents more annotated AI-generated images from FakeXplained. The left column displays the human annotations of FakeXplained. The right column shows the inference results of our best model, FakeXplainer. The center bar indicates the proportion of human preference votes from our user study. Note that since the "neutral" option was allowed, although the third annotated image received 46.2% of the votes, the human annotator is still rated higher than our model response. Our model demonstrates the ability to generate clearer, more descriptive captions for fake regions and reliably identifies content that contradicts common sense. For instance, in the lock-and-keyhole example (row 5), the model successfully detects that the key is not inserted into the correct keyhole. In the volcano example (row 2), in addition to identifying the "broken mountain body" as in the human annotation, the model also detects a subtle issue: the disconnection of the lava flow, highlighting its fine-grained visual reasoning capabilities.

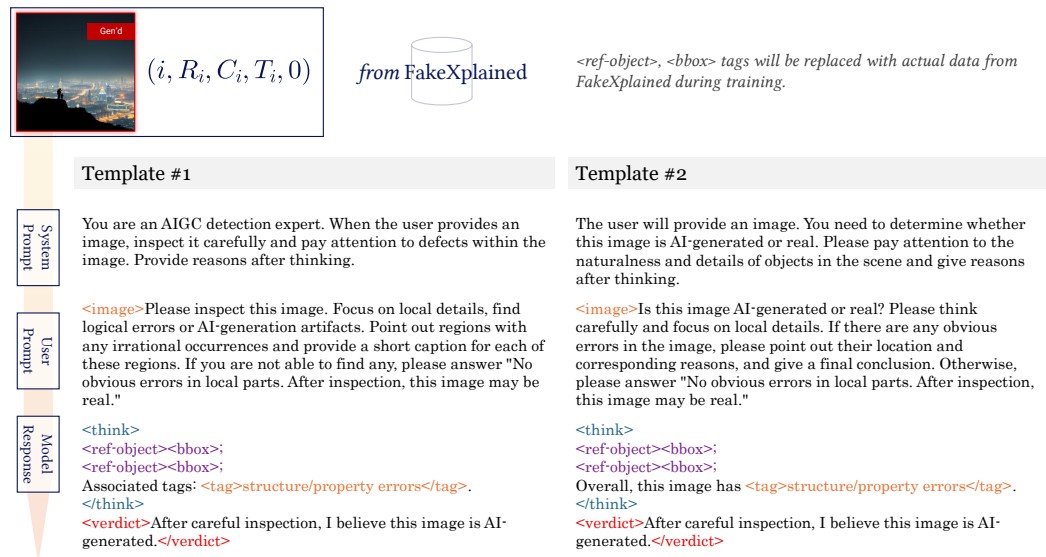

Figure 8: An example showing two different chat templates branched from one annotation entry.

## A.3 ETHICAL CONSIDERATIONS

All generated images are synthetic with no real individuals. Annotators provided informed consent to this annotation job, allowing us to use the annotated dataset for training. We explicitly ask the annotators not to leave any personal or sensitive information in annotations.

## A.4 KNOWN LIMITATIONS

**Language.** Currently, all annotations are in one language. It is hard to translate the short annotation sentences to other languages without a manual check for language inconsistencies.

**No Real Images.** FakeXplained does not contain real images for the time being, as defining regions for real images can be more subjective than AI-generated images.

## B ADDITIONAL TRAINING DETAILS

### B.1 TWO-STAGE TRAINING

We use ms-swift Zhao et al. (2025) for fine-tuning Qwen-2.5-VL models.

In the LoRA SFT stage, we noticed that freezing either the projector or the vision encoder leads to marginal improvement over the base model without training. To achieve optimal SFT performance, both modules must be fine-tuned jointly.

After the SFT stage, we use GRPO instead of PPO. As noted in Shao et al. (2024), GRPO obviates the need for additional value function approximation as in PPO, and instead uses the average reward of multiple sampled outputs. For each query $q$, GRPO samples $G$ outputs $\{o_1, o_2, \ldots, o_G\}$ from the old policy model $\pi_{\theta_{old}}$, and uses the relative *advantage* to optimize the MLLM, making it particularly well-suited for multi-modal reasoning tasks where absolute reward calibration is challenging.

We set the initial learning rate to $10^{-4}$ for the SFT stage and $10^{-5}$ for the RLHF stage. Reward signals fluctuated at the beginning of GRPO but quickly converged as the model is generating more human-aligned explanations, confirming the effectiveness of our reward design and training strategy.

### B.2 COMPUTATIONAL RESOURCES

The full training procedure took 41.0 hours on 8x NVIDIA A100 (80G) GPUs.

Table 5: Comparative performance analysis under compression artifacts, spatial transformations, and resolution changes.

| Degradation & Metric | | **FakeXplainer** | ObjFormer. | SegFormer | FakeShield | LEGION | NPR | DMD. | ComFor. | AfPr. | AEROB. | DIRE |
|---|---|---|---|---|---|---|---|---|---|---|---|---|
| JPEG Compression | Acc. | **0.979** | 0.940 | 0.927 | 0.782 | 0.544 | 0.820 | 0.908 | 0.840 | 0.871 | 0.842 | 0.884 |
| (80% Quality) | IoU | **0.353** | 0.284 | 0.231 | 0.092 | 0.061 | - | - | - | - | - | - |
| JPEG Compression | Acc. | **0.977** | 0.926 | 0.915 | 0.735 | 0.535 | 0.781 | 0.897 | 0.784 | 0.856 | 0.814 | 0.879 |
| (30% Quality) | IoU | **0.339** | 0.267 | 0.198 | 0.078 | 0.059 | - | - | - | - | - | - |
| Random Cropping | Acc. | **0.962** | 0.943 | 0.934 | 0.756 | 0.519 | 0.903 | 0.915 | 0.829 | 0.879 | 0.858 | 0.891 |
| | IoU | **0.314** | 0.217 | 0.176 | 0.067 | 0.061 | - | - | - | - | - | - |
| Downsampling | Acc. | **0.980** | 0.929 | 0.931 | 0.748 | 0.591 | 0.899 | 0.912 | 0.853 | 0.875 | 0.841 | 0.894 |
| (0.5x) | IoU | **0.362** | 0.259 | 0.254 | 0.092 | 0.076 | - | - | - | - | - | - |
| Original | Acc. | **0.982** | 0.954 | 0.945 | 0.801 | 0.583 | 0.914 | 0.928 | 0.882 | 0.887 | 0.873 | 0.911 |
| Images | IoU | **0.360** | 0.299 | 0.289 | 0.028 | 0.098 | - | - | - | - | - | - |

Table 6: Out-of-distribution performance evaluation across different datasets when trained with various configurations mentioned in the paper.

| Sources | **FakeXplainer** | No-FT | Partial Data | | | | Training Strategy | | |
|---|---|---|---|---|---|---|---|---|---|
| | | | no-bbox | no-caption | no-tags | label-only | SFT | GRPO$_{\omega_G=1}$ | GRPO$_{\omega_G=0.5}$ |
| GPT-Image-1 Rapidata (2025) | **0.801** | 0.421 | 0.691 | 0.760 | 0.774 | 0.603 | 0.591 | 0.788 | 0.768 |
| FaceForensics++ Rössler et al. (2019) | **0.864** | 0.519 | 0.715 | 0.796 | 0.817 | 0.640 | 0.680 | 0.826 | 0.832 |
| MMFR-Dataset Gao et al. (2025) | **0.874** | 0.593 | 0.859 | 0.708 | 0.843 | 0.612 | 0.671 | 0.794 | 0.773 |

At inference time, the end-to-end pipeline that takes an image as input to generate the verdict and grounding (if the image is deemed AI-generated) takes an average of 7.8 seconds on 2x NVIDIA A100 (80G) GPUs.

## C ROBUSTNESS AGAINST IMAGE PERTURBATIONS

To evaluate the practical applicability of our approach, we conduct a comprehensive robustness evaluation under common image degradations that are frequently encountered in real-world scenarios. Table 5 presents a comparative performance analysis across three perturbation categories: JPEG compression, random cropping, and downsampling.

Our method demonstrates exceptional resilience to JPEG compression artifacts, achieving low performance degradations of merely $0.3\%$ and $0.8\%$ from the uncompressed baseline, significantly outperforming current state-of-the-art methods. All of which experience at least a $3\%$ degradation. Notably, SegFormer and ObjectFormer show more stability than image-only classification models, indicating that grounding enhances robustness, although they still fall short of our method. For downsampling, we scaled the input images to 50% of their original width and height. In random cropping and downsampling experiments, our approach achieves the accuracy of 96.2% and 98.0%, respectively, indicating robust performance across different resolution scales. Meanwhile, downsampling does not severely affect the IoU score, which suggests that our grounded reasoning approach effectively captures semantic-level artifacts that remain detectable even at reduced resolutions, unlike methods that may rely on pixel-level features more susceptible to resolution changes. Since random cropping modifies the overall image layout, this action can remove certain fake regions from an image entirely, leading to lower IoU across all methods. Interestingly, we observe a slight increase in IoU after downsampling. We hypothesize that this is because our grounding model focuses on the dominant artifact region, which remains visible at lower resolutions, while noisy fine details are suppressed, leading to more precise and focused localization. Overall, the consistent performance across perturbation types demonstrates that our model captures underlying semantic artifacts in AI-generated content, enabling robust detection even under challenging image conditions.

Figure 9: More annotation examples from FakeXplained and model response visualized from FakeXplainer. The ratio in the center shows the human preference score.

Table 7: Performance analysis of FakeXplainer-32B across different image tags. The table shows the percentage of samples containing each tag, and the classification accuracy for images with (Acc on X) and without (Acc on non-X) that specific tag.

| Tag (X) | With X | Without X | Samples |
|---|---|---|---|
| Structure/Attribute Error | 98.39% | 97.43% | 80.52% |
| Wrong Texture | 98.53% | 97.95% | 42.65% |
| Artistic Style | **100.00%** | 97.95% | 12.18% |
| Other Anomalies | 94.08% | 98.45% | 5.78% |
| Unrecognizable Objects | 96.72% | 98.25% | 3.12% |
| Perspective Errors | **100.00%** | 98.18% | 1.16% |

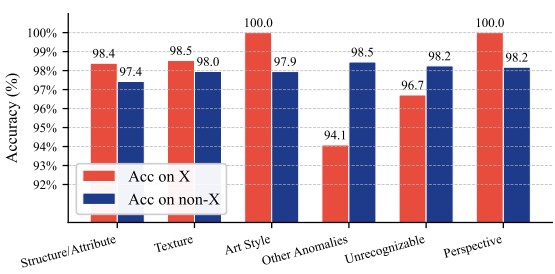

Figure 10: Visual representation of detection accuracy across different tag categories.

## D    ADDITIONAL ABLATION STUDIES

### D.1    OUT-OF-DISTRIBUTION PERFORMANCE

We evaluate the generalization capabilities of the ablation models in Section 5.2 (Table 2) of the main paper on three OoD datasets: images generated by GPT-Image-1 Rapidata (2025), FaceForensics++ Rössler et al. (2019) and MMFR-Dataset (eval) proposed by FakeReasoning Gao et al. (2025). Table 6 shows that our complete pipeline achieves accuracies of 80.1, 86.4 and 87.4 respectively, compared to 42.1, 51.9 and 59.3 for the base model without fine-tuning.

Among partial data ablations, the label-only configuration performs the worst among all partial data category entries, yielding near no-finetuning performance. This OoD evaluation further confirms that both spatial grounding and textual reasoning are essential for generalization.

The SFT stage alone yields moderate performance (59.1 on GPT-Image-1, 68.0 on FF++, 67.1 on MMFR). Further into the GRPO training, we see a better overall performance. This result is consistent with findings discussed in our main paper, as the RLHF stages give more performance boost than the SFT stage. The consistent improvements across both datasets suggest our approach learns generalizable features for AI-generated content detection rather than dataset-specific patterns.

### D.2    DISABLING LoRA

We employ LoRA during training to reduce computational cost and memory usage. While full-parameter fine-tuning is technically possible, our results show that it does not improve accuracy or IoU (Accuracy: 98.2% → 97.9%, IoU: 36.0% → 35.4%), likely due to the limited amount of annotated data. This suggests that LoRA provides a more efficient and suitable training strategy under current data constraints. With significantly more training data, full fine-tuning may yield better results.

## E    DISCUSSIONS ON TAG-WISE PERFORMANCE

To assess the impact of different artifact types on detection performance, we analyzed the accuracy of FakeXplainer-32B on the FakeXplained dataset across the six distinct image-level tags defined in Section 3. Table 7 details the prevalence of each tag within the dataset and compares the model's accuracy on images containing a specific tag versus those without it.

Notably, the model achieves 100% accuracy on images tagged with **Artistic Style** and **Perspective Errors**. While Perspective Errors are rare (1.16%), the perfect detection rate implies that vanishing point inconsistencies are distinct features that the MLLM can easily leverage for classification. Meanwhile, the model exhibits a slight performance drop on images labeled with the **Other Anomalies** tag, which is typically assigned to images with global inconsistencies or subtle defects that are difficult to categorize into specific localized regions. This decrease in performance suggests that localized flaws are powerful cues to draw the conclusion that an image is AI-generated. This can potentially be improved with a global explanation to images in the FakeXplained dataset. Similarly,

**Unrecognizable Objects** (96.72%) presents a moderate challenge, likely because defining "recognizability" can be subjective and occasionally overlaps with abstract artistic intent.

## F    LIMITATIONS

Despite promising results, our approach still has limitations. The Qwen-2.5-VL-32B-Instruct model incurs substantial computational costs, which may limit its deployment in resource-constrained environments. Our evaluation does not sufficiently cover domain-specific or real-world image types, such as medical, industrial, or artistic imagery. As generated media becomes more and more realistic and cross-modal, FakeXplainer, designed to detect visible, human-interpretable artifacts that can be localized and explained, may not be able to produce explainable responses. This scope is inherent to all explainable AIGI detectors: grounded, human-aligned reasoning is only feasible when the underlying cues are perceptible to humans. However, as long as visible artifacts are still present, FakeXplainer can effectively detect and explain. Nonetheless, as long as perceptible cues remain, FakeXplainer provides significantly more robust and verifiable predictions than black-box classifiers, especially under distribution shift, due to its grounded and human-aligned reasoning process.

## G    BROADER IMPACT

While our system improves interpretability in detecting AI-generated content, it may also introduce risks. The detailed explanations of detection rationale could inadvertently assist malicious adversaries in developing more sophisticated evasion techniques, potentially contributing to an adversarial "arms race." The deployment of such systems without careful consideration could lead to over-censorship of legitimate content, particularly affecting artists and creators who use AI tools ethically. To mitigate these risks, we recommend responsible deployment frameworks, ongoing monitoring for bias and fairness, and collaborative development with stakeholders to ensure the technology serves the public interest while preserving legitimate creative expression.

## H    THE USE OF LARGE LANGUAGE MODELS

During manuscript preparation, we employed LLMs only for language polishing and grammar refinement. All research ideas, methods, and results were conceived, implemented, and validated entirely by the authors. Since our work studies MLLMs in the context of forgery detection, we necessarily employed LLMs as research subjects. Specifically, MLLMs were used to generate or assist in generating annotations within our dataset and to serve as baseline models in our experiments. These usages are intrinsic to the research problem itself and should not be interpreted as LLMs contributing to the ideation or authorship of this paper.

