# OpenReview forum: "FakeXplain: AI-Generated Image Detection via Human-Aligned Grounded Reasoning"
_ICLR.cc/2026/Conference — ICLR 2026 Poster_

### Official Review · Reviewer_N3fb · 2025-10-29

**Soundness:** 3
**Presentation:** 3
**Contribution:** 3
**Rating:** 6
**Confidence:** 4

**Summary:**

This paper presents FakeXplain, an artifact detection, localization, and explanation method. To train the FakeXplainer, they first curate a synthetic dataset with region and textual annotations for artifacts (FakeXplained). Finetuning MLLMs on the FakeXplained dataset improves the detection and localization accuracy, as well as robustness and generalization on out-of-distribution cases.

**Strengths:**

- This paper studies an interesting task in artifact detection, localization, and explanation. They notice that the existing methods or datasets face challenges such as hallucination and low human alignment. They aim to build a reliable and interpretable MLLM-based system.
- They have solid contributions, including curating a dataset with 28 T2I models and using it to achieve a big improvement across different MLLMs.
- Extensive experiments and analysis are provided.

**Weaknesses:**

- Some important technical details are unclear, such as the image generation and filtering process.
- For the Table 1 experiment, it is unclear what benchmark they use. If they are using their own test split, then it is less impressive. Additionally, all the baseline methods have very good accuracy, which makes it seem doubtful how challenging the benchmark is.

**Questions:**

- For the image generation, how many images are generated by each model? How many images per model for each prompt? What's the filtering criteria and rate?
- For tag annotation, could you show the distribution of the tags (diversity, and distribution train/val/test)? When dividing data for SFT and RL, do you use the tag to control difficulty?
- What is the benchmark you use for Table 1? Could you show the correlation between accuracy and tags? In what category does your method perform better, and how do SFT and RL improve the performance distinctively?

---

> ### Author Response · Authors · 2025-11-19
> **Rebuttal to Reviewer N3fb**
>
> We sincerely thank the reviewer `N3fb` for acknowledging our solid dataset contribution and extensive empirical analysis.
>
> > RQ1/RW1. How many images are generated by each model? How many images per model for each prompt? What's the filtering criteria and rate?
>
> We generate 1,000 images per model, one image per ImageNet-1K class. We only use one prompt throughout the generation process: "A realistic image of <class_name>."
>
> We manually filter out low-quality images that depict non-identifiable objects or correspond to no plausible real-world scene.
>
> Thank you for the suggestions. *Appendix A.1* is now updated to clarify these details.
>
> > RQ2. For tag annotation, could you show the distribution of the tags (diversity, and distribution train/val/test)? When dividing data for SFT and RL, do you use the tag to control difficulty?
>
> The distribution (occurrences) of the tags is provided in the response to RQ3/RW2a below.
>
> Importantly, tags in FakeXplained serve as an auxiliary component of explainability. They summarize the semantic type of artifact described in the bounding-box captions, but they are not used as primary supervision for grounding.
> We therefore do not use the tags to control the difficulty for the train-test split. Both the SFT and RL subsets are uniformly sampled from the training split to preserve the natural distribution of artifacts and avoid biasing the optimization process.
> Tags are used only to support human-aligned explanation quality and later diagnostic analysis, rather than to influence the model’s training schedule.
>
> > RQ3/RW2a. What is the benchmark you use for Table 1? Could you show the correlation between accuracy and tags? In what category does your method perform better, and how do SFT and RL improve the performance distinctively?
>
> Table 1 reports four-fold cross-validation on FakeXplained: in each fold, we train on 75% of the AI-generated images plus an equal number of real images, and evaluate on the remaining 25% of AI images and the matched real ones.
>
> The table below shows the performance of FakeXplainer-32B on FakeXplained. While tags do not significantly influence the model's performance, it is worth noting that the model excels at identifying AI-generated images with artistic style or perspective errors, but struggles with images labeled with the `Other Anomalies` tag. This can be caused by the fact that images with the `Other Anomalies` tag tend to have global issues that are hard to localize.
>
> | X                         | Acc on X | Acc on non-X | Samples |
> | ------------------------- | -------- | ------------ | ------- |
> | Wrong Texture             | 98.53%   | 97.95%       | 42.65%  |
> | Structure/Attribute Error | 98.39%   | 97.43%       | 80.52%  |
> | Art Style                 | 100.00%  | 97.95%       | 12.18%  |
> | Unrecognizable Objects    | 96.72%   | 98.25%       | 3.12%   |
> | Perspective Errors        | 100.00%  | 98.18%       | 1.16%   |
> | Other Anomalies           | 94.08%   | 98.45%       | 5.78%   |
>
> The table and the corresponding discussions are now reported in a dedicated section (*Appendix E*) of the revised paper. We sincerely thank the reviewer for this insightful advice.
>
> > RW2b. All the baseline methods have very good accuracy, which makes it seem doubtful how challenging the benchmark is.
>
> All baselines in Table 1 are trained on the FakeXplained training split to ensure a fair comparison, as we mentioned at the beginning of Sec. 5.2.
> This in-distribution setup is strong for all methods, which explains why several baselines exceed 90% accuracy, an observation consistent with prior AIGI benchmarks such as [1-3].
> Importantly, this does *not* imply that the benchmark is trivial.
> When baselines are evaluated without training on FakeXplained, their performance drops substantially (*e.g.*, the strongest baseline, NPR, reaches only 70.42% accuracy).
> This indicates that FakeXplained provides non-trivial supervision signals that other datasets do not offer.
> Furthermore, Tables 3 demonstrate that the combination of our dataset and SFT+GRPO pipeline yields significantly stronger *out-of-distribution generalization*: FakeXplainer outperforms all baselines on GPT-Image-1, FF++, and MMFR.
>
> Our primary focus is grounded, human-aligned explainability, where accuracy alone is insufficient. FakeXplainer achieves substantially higher IoU than SegFormer and ObjectFormer, and our human-preference study shows near-parity with human annotators. Additionally, it surpasses existing MLLM-based detectors in BLEU-2, ROUGE-L, and detection accuracy. These results confirm that FakeXplained introduces a challenging and informative benchmark for models that must both detect artifacts and justify their reasoning in a human-aligned manner.
>
> [1] A Million-Scale Benchmark for Detecting AI-Generated Image. NeurIPS, 2023.
>
> [2] A Sanity Check for AI-generated Image Detection. ICLR, 2025.
>
> [3] FakeBench: Probing Explainable Fake Image Detection via Large Multimodal Models. TIFS 2025.

---

### Official Review · Reviewer_iUyJ · 2025-10-31

**Soundness:** 3
**Presentation:** 4
**Contribution:** 3
**Rating:** 6
**Confidence:** 4

**Summary:**

This paper introduces FakeXplained, a new dataset for explainable AI-generated image (AIGI) detection, featuring human-annotated bounding boxes and captions for synthesis artifacts. The authors use this to train FakeXplainer, a model based on an SFT+RLHF (GRPO) pipeline, which achieves high detection accuracy (98.2%) and provides human-aligned explanations.

**Strengths:**

1. Valuable Dataset: The FakeXplained dataset is a strong contribution, enabling the development and benchmarking of explainable AIGI detectors.
2. Strong Empirical Results: The model achieves SOTA accuracy (98.2%). The human preference study, showing near-parity with human annotators, is a compelling validation of the "human-alignment" claim.

**Weaknesses:**

1. Limited Methodological Novelty: The paper's methodological contribution is somewhat limited. The core SFT+GRPO training pipeline is a successful application and adaptation of existing frameworks (e.g., from DeepSeek-Math) rather than a new algorithm.
2. Inherent Limitation of the Task Formulation: The paper's core premise, "Human-Aligned Grounded Reasoning," focuses by definition on human-perceptible flaws. This is an inherent challenge for all current work in explainable AIGI detection, as it does not address the separate problem of detecting SOTA images that may be "fake" but lack obvious, semantically explainable artifacts.
3. Potential for Overfitting: There is a notable mismatch between the modest dataset size (8,772 images) and the large models being fine-tuned (32B parameters). This poses a risk of overfitting.
4. High Computational Cost: The SOTA performance relies on a 32B parameter model, which, as the authors note, incurs substantial computational costs. The reported inference time of 7.8 seconds per image (on 2x A100 GPUs) and 41 hours of training (on 8x A100 GPUs) limits the method's practical applicability in real-time or resource-constrained environments.

**Questions:**

1. Since the SFT+GRPO pipeline is adapted from existing work, what is the specific, non-trivial novelty introduced in the model structure or training details that makes this method unique for AIGI detection?
2. The authors should explicitly discuss the limitation mentioned in Weakness #2 in their conclusion. Clarifying that the scope of this work is "detecting visible, explainable artifacts" rather than all AIGI, would strengthen the paper's positioning.
3. To improve reproducibility, the authors should report the specific values for all critical hyperparameters, such as the batch size, reward relaxing constant $\eta$ and the GRPO sampling number $G$.

---

> ### Author Response · Authors · 2025-11-19
> **Rebuttal to Reviewer iUvJ**
>
> We sincerely appreciate Reviewer `iUyJ` for acknowledging the value of our dataset and the strong empirical results.
>
> > RQ1/RW1. Since the SFT+GRPO pipeline is adapted from existing work, what is the specific, non-trivial novelty introduced in the model structure or training details that makes this method unique for AIGI detection?
>
> Although our method builds on the general SFT+GRPO paradigm, our contribution is an AIGI-specific training framework that equips a generic MLLM with human-aligned detection, spatial grounding, and explanation, which prior GRPO applications (*e.g.*, math, text reasoning) do not address.
>
> - **Reward design.** We introduce a three-part reward tailored to AIGI detection: (1) classification correctness, (2) IoU-based grounding of artifact regions, and (3) strict adherence to our structured output format. This reward tightly couples the model’s learning objective with how humans justify decisions, making the training process human-aligned by design.
> - Progressive GRPO (**p**GRPO). We propose a dynamic grounding weight schedule that progressively increases the localization reward during GRPO. Our ablation studies show that this mechanism is crucial for avoiding degenerate box predictions, stabilizing training, and improving both IoU and human-preference scores over SFT-only and fixed-weight RL variants.
>
> Together, these components form a non-trivial, AIGI-specific extension of the SFT+GRPO paradigm. They operationalize reliable, spatially grounded reasoning, a capability absent in prior GRPO works, and are key to enabling the human-aligned behavior required by our task.
>
> > RQ2/RW2. The authors should explicitly discuss the limitation mentioned in Weakness #2 in their conclusion. Clarifying that the scope of this work is "detecting visible, explainable artifacts" rather than all AIGI, would strengthen the paper's positioning.
>
> Our method is designed for cases where AI-generated images contain visible, *human-interpretable artifacts* that can be localized and explained. This scope is inherent to *all* explainable AIGI detectors: grounded, human-aligned reasoning is only feasible when the underlying cues are perceptible to humans.
> This is therefore methodological rather than specific to our implementation.
> In practice, current generators still exhibit subtle but observable irregularities in many cases, and FakeXplainer is specifically built to detect and explain such cues.
> Moreover, our experiments show that FakeXplainer achieves **competitive or superior accuracy and robustness** compared with several black-box AIGI detectors on external benchmarks (*e.g.*, GPT-Image-1, FF++, MMFR in Table 3), indicating that human-aligned, spatially grounded reasoning does not come at the cost of predictive performance.
> Thank you for the suggestion, and we have added relevant discussion to the *Conclusions section (Sec. 6)* in the revised paper.
>
> > RQ3/RW3. To improve reproducibility, the authors should report the specific values for all critical hyperparameters, such as the batch size, reward relaxing constant and the GRPO sampling number.
>
> We appreciate this observation, and the key hyperparameters will be added to Section 5.1 of the revised manuscript.
> Both the SFT and GRPO stages last for three epochs with a batch size of 1. We use $\eta=1.1$ and set the number of samples to $G=4$ during the GRPO stage.
> We will release the full training code with the final version of the paper.
>
> > RW4. High Computational Cost.
>
> The FakeXplainer-32B model in our experiments serves primarily as a performance *upper bound*, not a deployment requirement. As shown in our scaling study (Table 4), *the 7B variant already achieves 95.8% accuracy with strong spatial grounding, and even the 3B model provides usable performance*. This demonstrates that FakeXplainer can be deployed on smaller architectures with limited degradation.
>
> As mentioned by the reviewer, our flagship FakeXplainer-32B has an average end-to-end inference time of **7.8s per image**. Our 7B variant reaches **3.8s** per image, which is considerably more efficient than existing explainable baselines such as *LEGION (13.7s)* and *FakeShield (30.6s)* on the same hardware, while providing more coherent and accurate explanations.

---

### Official Review · Reviewer_bATs · 2025-11-01

**Soundness:** 3
**Presentation:** 3
**Contribution:** 3
**Rating:** 6
**Confidence:** 4

**Summary:**

This paper addresses the critical challenges of interpretability and generalization in AI-generated image detection. The authors argue that existing methods are often "black boxes" that fail to generalize to out-of-distribution (OOD) data, while multi-modal large language models (MLLMs) tend to "hallucinate" explanations lacking reliable visual grounding. To overcome this, the paper introduces two key contributions: 1) The FakeXplained dataset, a novel collection of AI-generated images meticulously annotated by humans with bounding boxes and descriptive captions that pinpoint specific synthesis artifacts, providing a foundation for human-aligned, grounded reasoning. 2) The FakeXplainer method, a detector developed by fine-tuning an MLLM on this dataset using a progressive training pipeline (SFT + GRPO). Experiments demonstrate that FakeXplainer achieves state-of-the-art performance in both detection accuracy (98.2%) and artifact localization IoU (36.0%), while also showing strong OOD generalization and robustness. Its primary advantage is the ability to generate spatially grounded, human-aligned textual explanations for its verdicts, effectively mitigating the hallucination issues common in MLLMs.

**Strengths:**

The method provides trustworthy, spatially grounded textual explanations for AI detection, mitigating MLLM hallucination.

The paper introduces the FakeXplained dataset, featuring meticulous human annotations (bounding boxes and captions) essential for training reliable, explainable models.

**Weaknesses:**

The model shows strong performance on the current dataset. To further validate its generalization and mitigate potential overfitting risks, it would be beneficial to test its zero-shot capabilities on an external dataset, such as Chamelon[1].

The analysis could be further strengthened by discussing and comparing against other relevant explainable MLLM-based detectors, for instance, FakeVLM[2], So-Fake[3], and FakeScope[4].

To provide a more comprehensive picture of the method's capabilities, it would be valuable to extend the evaluation to include additional testable benchmarks, such as Fakebench[5] and LOKI[6].

[1] A Sanity Check for AI-generated Image Detection.

[2] Spot the fake: Large multimodal model-based synthetic image detection with artifact explanation.

[3] So-Fake: Benchmarking and Explaining Social Media Image Forgery Detection

[4] Fakescope: Large multimodal expert model for transparent ai-generated image

[5] FakeBench: Probing Explainable Fake Image Detection via Large Multimodal Models

[6] Loki: A comprehensive synthetic data detection benchmark using large multimodal models

**Questions:**

None

---

> ### Author Response · Authors · 2025-11-20
> **Rebuttal to Reviewer bATs**
>
> We sincerely thank Reviewer `bATs` for highlighting highlighting our human-aligned grounded explanations and the value of the FakeXplained dataset.
>
> For completeness, we discuss each referenced method:
>
> - Chameleon [1]: is an image-only benchmark for AI-generated image detection, without textual or region-level annotations. The original AIDE detector operates purely on visual features (CLIP embeddings and frequency patches), without any MLLM-based explanations. In our setting, we only use Chameleon as an out-of-distribution image-level detection benchmark.
> - FakeClue / FakeVLM [2]: FakeVLM is trained on FakeClue using fine-grained natural-language artifact descriptions, but the dataset does not provide explicit region-level bounding boxes or masks. We therefore regard its supervision as language-only rather than spatially grounded. In our experiments, we treat FakeClue as an out-of-distribution benchmark and include FakeVLM as an MLLM-based baseline.
> - So-Fake [3]: The manipulated regions are synthesized and weakly labeled using a segmentation and an inpainting model, rather than human-annotated or intrinsically grounded MLLM regions.
> - FakeScope [4]: is an expert LMM for AI-generated image forensics trained on FakeChain and FakeInstruct, which provide linguistic authenticity reasoning and multimodal instructions but no explicit region-level annotations. FakeScope focuses on detection and rich natural-language explanations rather than supervised region grounding.
> - FakeBench [5]: provides textual authenticity descriptions collected via a human-in-the-loop process based on a fine-grained forgery taxonomy, but does not release region-level annotations such as bounding boxes or masks. It is therefore suited to evaluating explainable fake-image detection at the image and text level rather than training grounded detectors.
> - LOKI [6]: is a multi-modal QA benchmark (video, image, 3D, audio, text) with both coarse-grained judgement / multiple-choice tasks and fine-grained anomaly-selection / explanation tasks. For images and 3D, fine-grained anomaly regions are manually annotated using tools such as LabelU, while GPT-4o is mainly used to generate or refine textual questions and abnormal-detail descriptions on top of these human-labeled anomalies. As a result, LOKI is primarily designed to evaluate LMM QA and anomaly reasoning, rather than to serve as a large-scale human-annotated training set for grounded detection.
>
> Across these works, *none offer fully human-annotated region-level grounding*, and several do not release code or annotations. This highlights the need for a dataset like FakeXplained that provides **explicit human spatial groundings and human-aligned explanations**, enabling consistent evaluation of grounded reasoning.
>
> Considering publicly released models and data, we incorporate: Chameleon [1] and FakeClue [2] as ood benchmarks and FakeVLM [2] as an MLLM-based baseline.
>
> *We have 1) cited all the mentioned papers [1-6] and discuss them in the Related Works section, 2) reported the zero-shot OoD accuracy on Chameleon [1] and FakeClue [2] in Table 3 and 3) reported the evaluation result of FakeVLM [2] in Table 1.*
>
> A summary of the added experimental results are shown below:
>
> |  Sources  | FXP.  | ObjFormer. | SegFormer |  NPR  | DMD.  | ComFor. | AfPr. | AEROB. | DIRE  | FakeShield | LEGION |
> |:---------:|:-----:|:----------:|:---------:|:-----:|:-----:|:-------:|:-----:|:------:|:-----:|:----------:|:------:|
> | FakeClue  | **0.852** |   0.462    |   0.485   | 0.833 | 0.734 |  0.766  | 0.849 | 0.239  | 0.727 |   0.550    | 0.172  |
> | Chameleon | **0.843** |   0.485    |   0.508   | 0.794 | 0.721 |  0.757  | 0.803 | 0.291  | 0.752 |   0.587    | 0.197  |
>
> The table above shows the performance of our method and baseline methods on this OoD dataset. FakeXplainer (FXP.) demonstrates outstanding performance against all evaluated baseline methods.
>
> To broaden comparisons with explainable MLLM-based detectors, we fine-tuned FakeVLM [2] on FakeXplained as a plain MLLM baseline. As shown below,
>
> |  | Diffusion | GAN   | DiT   | Others | Real  | Overall |
> | :----------- | :-------- | :---- | :---- | :----- | :---- | :------ |
> | FakeVLM      | 0.919     | 0.827 | 0.892 | 0.870  | 0.763 | 0.828   |
> | FakeXplainer | 0.983     | 0.955 | 0.983 | 0.978  | 0.985 | **0.982**   |
>
> FakeXplainer achieves higher accuracy across all categories, confirming its clear advantage over FakeVLM.
>
> [1] A Sanity Check for AI-generated Image Detection
>
> [2] Spot the Fake: Large Multimodal Model-based Synthetic Image Detection with Artifact Explanation
>
> [3] So-Fake: Benchmarking and Explaining Social Media Image Forgery Detection
>
> [4] Fakescope: Large Multimodal Expert Model for Transparent AI-generated Image Detection
>
> [5] FakeBench: Probing Explainable Fake Image Detection via Large Multimodal Models
>
> [6] LOKI: A Comprehensive Synthetic Data Detection Benchmark Using Large Multimodal Models

---

> > ### Comment · Reviewer_bATs · 2025-11-27
> >
> > My questions have been largely answered, and I have decided to keep my score unchanged.

---

### Official Review · Reviewer_gGWh · 2025-11-04

**Soundness:** 3
**Presentation:** 3
**Contribution:** 3
**Rating:** 8
**Confidence:** 3

**Summary:**

The paper introduces FakeXplained, a dataset of 8,772 AI-generated images annotated with region-level bounding boxes and short captions (≈5.42 per image), as well as image-level tags, and FakeXplainer, a progressive SFT→RLHF (pGRPO) fine-tuning pipeline for MLLMs that detects, localizes, and explains synthesis artifacts. The method optimizes three rewards—classification, grounding, and format. Experiments demonstrate that the method yields strong results: overall 98.2% accuracy and 36.0% IoU on their splits, with ablations showing large gains over SFT-only and fixed-weight RL.

**Strengths:**

- Clear problem framing + resources. A well-specified dataset, an expert-guided annotation protocol (23 annotators), and concrete artifacts improve supervision for grounding and reasoning.
- Strong empirical results. High in-distribution accuracy/IoU, consistent ablation results (e.g., captions/boxes matter), and human-preference-parity trends support the claim of explainable detection.
- Method integrates structures and learning. The <think>/<tag>/<verdict> output format aligns neatly with reward design, reducing formatting errors and encouraging grounded explanations.

**Weaknesses:**

- Computational demands. Best results rely on large MLLMs (e.g., Qwen-2.5-VL-32B) with noted deployment costs.
- Lenient quality-control thresholds may admit noisy labels. The QC accepts region matches at just 20% IoU and image-level tag accuracy of 1/3, which risks training/evaluating on imprecise boxes and weak tag agreements—particularly harmful for grounding-heavy RL.

**Questions:**

None.

---

> ### Author Response · Authors · 2025-11-19
> **Rebuttal to Reviewer gGWh**
>
> We sincerely thank the reviewer `gGWh` for highlighting the strength of our empirical results and the effectiveness of our progressive fine-tuning pipeline.
>
> > RW1. Computational demands. Best results rely on large MLLMs (e.g., Qwen-2.5-VL-32B) with noted deployment costs.
>
> In our experiments, the 32B FakeXplainer model serves primarily as a performance *upper bound*, not a deployment requirement. As shown in our scaling study (Table 4), the **7B variant already achieves 95.8% accuracy** with strong spatial grounding, and even the 3B model provides usable performance. This demonstrates that FakeXplainer can be deployed on smaller architectures with limited degradation.
>
> The additional inference cost mainly arises from generating *spatially grounded, human-aligned explanations*, which are essential for trustworthy AIGI detection in high-stakes scenarios. In terms of wall-clock latency, our end-to-end inference time is **7.8s (32B) and 3.8s (7B)** per image, already **faster** than existing MLLM-based baselines such as *LEGION (13.7s) and FakeShield (30.6s)* on the same hardware, while providing more coherent and accurate explanations.
>
> > RW2. Lenient quality-control thresholds may admit noisy labels. The QC accepts region matches at just 20% IoU and image-level tag accuracy of 1/3, which risks training/evaluating on imprecise boxes and weak tag agreements—particularly harmful for grounding-heavy RL.
>
> Thank you for the insightful comment. We clarify that the 20% IoU and 1/3 tag-agreement thresholds are used *only for annotator quality control* (QC), **not for training or evaluation**. These thresholds serve to *remove clearly incorrect annotations while accommodating variability across annotators* in a task where valid artifact cues often occupy extremely small regions (*e.g.*, an extra finger or a distorted toenail). In such cases, both a tight box around the flaw and a slightly larger box covering the full hand remain semantically valid as long as the causal region is included and the associated caption is reasonable.
>
> In the final FakeXplained dataset, the annotations reached an average IoU of *42.35%*, and the final tag-agreement rate is *79.67%* (evaluated against the QC samples). These metrics indicate that FakeXplained provides strong overall consistency rather than weak supervision.
>
> We have added a new paragraph in *Appendix A.2* of the revised paper to discuss the details of the QC process. *Figure 7* now provides representative samples to qualitatively demonstrate the 20\% IoU requirement.

---

### Meta-Review · Area_Chair_Sdhx · 2026-01-06

**Summary:**

The paper proposes a very interesting dataset for AI generated content detection where the unique thing in the dataset is the explainability part. This is a very important contribution to the field of AI generated data detection.

**Reviewer Concerns:**

It seems that the authors addressed all the concerns in the rebuttal

**Reviewer Scores:**

All reviewers started with high scores and the authors answered all the questions raised so I believe the reviewers at least would not reduce their scores so the paper should get accepted. The 8 would probably remain 8. The 6 have the potential to increase to 7 as answers were provided to the questions raises (one explicitly mentioned that the 6 will remain unchanged)

---

### Decision · Program_Chairs · 2026-01-26

Accept (Poster)